# Expert decision-making in clinicians: An auto-analytic ethnographic study of operational decision-making in urgent care

**Nicola Irvine**[1]*, **Robert Van Der Meer**[2], **Itamar Megiddo**[2]

**1** Yunus Centre for Social Business and Health, Glasgow Caledonian University, Glasgow, Scotland,
**2** Department of Management Science, Strathclyde Business School, University of Strathclyde, Glasgow, Scotland

* nicola.irvine@gcu.ac.uk

**Data Availability Statement:** All relevant data are within the manuscript and its SupportingInformation files. The manuscript also contains the address and hyperlink link to an open

## Abstract

### Objective

To conceptualise the cognitive processes of early expert decision-making in urgent care.

### Background

Expert clinicians in the UK frequently determine suitable urgent care patient pathways via telephone triage. This strategy is promoted by policymakers but how it is performed, and its effectiveness has not been evaluated. Evaluation of early senior decision-making requires knowledge of decision-processes, influences, and goals. Previous research has focused on diagnostic decision-making and rarely studied clinicians in the field.

### Method

We triangulated analytic autoethnography of early expert decision-making with focused ethnography of experts and trainee doctors performing the task. The study took place in a medium-sized Acute Medical Unit which provided internal medical emergency care for a mixed urban and rural population in the UK. A grounded theoretical model of expert decision-making was created via Gioia Methodology. Decision types were categorised to identify differences in solutions as well as decision processes.

### Results

The hallmarks of intuitive decision-making were found in most expert decisions. Experts made intuitive use of pattern-matching to extract key data from large volumes of information which triggered the spontaneous manifestation of solutions. Solutions were holistic and usually solitary. Upon manifestation solutions were consciously tested for viability with emotional affect playing a key role. Expert solutions were previously applied ones but were frequently entirely novel. Novel solution generation was not a feature of trainee decisions but moments of intuition were. Expert goals varied between optimal care for individual

access repository held at the university of
strathclyde with additional data accessible at DOI:
10.15129/6af9033c-b8bc-4629-93ec-
e4fc8c921611.

**Funding:** The author(s) received no specific
funding for this work.

**Competing interests:** No competing interest.

patients, system-wide efficiency, and equity of care. The decision environment had a large
influence upon experts.

## Conclusion

Expert clinicians employ intuitive decision-making supported by rational analysis in early
urgent care decision-making. Expert solutions generated in this manner are pragmatic
rather than optimal, context dependent, and seek to achieve goals which vary from
moment-moment. Findings are crucial to inform research evaluating the effectiveness of
early expert decision-making in urgent care as it is a high cost strategy. They also have
implications for methodological approaches in future studies of expert clinical decision-mak-
ing, developing artificial expert systems, and clinician training.

## Introduction

Healthcare leaders seek ways to reduce demand upon in-patient resources by providing urgent
care via out-patient facilities [1–4]. Recognition of a patient's suitability for non-admission
when urgently unwell requires a degree of clinical expertise [5]. In the UK, policymakers and
healthcare leaders recommend telephone triage performed by senior clinicians at the moment
a patient is referred into hospital [6–8]. Termed early senior decision-making (ESDM), it is an
approach reported outside of UK settings too [9]. Whilst in principle this strategy has merits,
there is currently little evidence to support policymaker assumptions that the operational deci-
sion events of ESDM realise value in terms of patient outcomes and system efficiency when
employing senior doctors (consultants) in this role [10,11]. That said, the tendency of early
expert decisions to realise efficiency in urgent care compared with other staff decision-making
suggests that knowledge of how these decision events are performed would be useful for plan-
ning and developing services and clinical training [8,12].

Despite advocacy of ESDM in UK healthcare policy across all facets of urgent care [6], we
know little of how senior clinicians navigate conflicting/absent clinical evidence, the influence
of external factors, nor how operational decisions differ from diagnostic ones [13–16]. Our
research sought to address this gap via an auto-analytical ethnographic study of ESDM trian-
gulated via ethnographic observation of ESDM in action. We sought to contribute to existing
knowledge of expert decision-making by observing a group of professionals that are rarely
studied in the field. We begin with a summary of current knowledge of decision-making in cli-
nicians before presenting our findings, and then discussing how they contribute to literature
and where findings may be applied in future studies.

### Current knowledge

Contrary to historical models of conscious, logical deduction [17], clinicians have now been
shown to utilise conscious and non-conscious cognitive processes simultaneously during diag-
nostic decision-making [18]. These models have evolved from purely hypotheticodeductive
reasoning to incorporate psychology and behavioural economics evidence [17–21]. However,
research has been limited to diagnostic decision-making with few studies on the decisions
faced by clinicians in real world settings [15].

In experimental studies, both expert and non-expert clinicians form hypotheses early in
decision-tasks [17,19], often before all data is available [19,20]. This indicates the use of rapid,

Table 1. Manifestations of systems thinking.

| Mode | Origin | Manifestations |
|------|--------|----------------|
| Fast thinking (system one) | Non-conscious brain | **Guesswork** –a random decision without apparent influence<br>**Instinct** –decision via a fixed, hard-wired behaviour in response to stimuli (e.g., fear). Emotion as a driver of decisions<br>**Heuristics**–decision via a low effort, non-holistic mental short-cut. Emotion not a feature<br>**Intuition** –decision via non-sequential processing of internal and external information that requires little mental effort, is holistic, and associated with an emergent emotional change (e.g., sensation of relief) |
| Slow thinking (system two) | Conscious brain | **Rational analysis** - comparison of known (or perceived) costs and consequences of alternative decision outcomes |

[21,22,26–28].

non-conscious cognitive processes, regardless of expertise [22]. However, expert clinicians outperform non-experts in decision-making accuracy [19].

Distinguishing between expert and non-expert cognitive processes is challenging [23–25]. Referred to as 'system one' (or fast thinking), non-conscious decision-events may be as logical as consciously deliberated ones ('system two' or slow thinking), but the speed at which they occur leaves this unclear [25]. There are several different forms of fast thinking thought to be useful in different contexts (Table 1). For example, both heuristics and intuitive decisions apply prior knowledge and mental short-cuts to reduce cognitive burden but do so in different ways and with different results [22–25].

Studies often overlook differences between heuristics and intuitive decision-making among clinicians. Further, they are methodologically focused on the negative impact of heuristics with limited representativeness of expertise amongst study participants [29]. Field studies are rare. This leaves gaps in our knowledge of how expert clinicians usefully employ fast thinking in real settings, particularly for decisions outside of diagnosing [15].

## Methodology

A methodological bricolage combining a focused ethnographic case study with analytic auto-ethnography was used to study decision-making in consultants and trainees delivering acute (adult) medical care [30–34]. Analytic autoethnography (AA) is an ethnographic methodology that triangulates the reflexive findings of self (autoethnography) with the observations and reflections of others experiencing the same phenomenon (ethnography) [30]. Reflexive analysis of self is aided by Bracketing–a process where the initial conclusions (e.g., about causal mechanisms) are placed aside whilst the researcher considers all possible other explanations before concluding [35]. Triangulation is enhanced via analysis of additional relevant material where available (e.g., email correspondence, meeting notes). AA transcends either autoethnography or ethnographic observation alone as a richer understanding of phenomena emerge from an analysis that moves iteratively between the self, others, and additional materials [36–38].

An AA approach exploited the dual role of our lead author as researcher in decision science and a consultant in acute internal medicine [30,39].

It exposed many of the underlying mechanisms and causal powers behind the ESDM task as a lived experience of clinicians allowing us to generate a conceptual theory of expert decision-making in the ESDM task [30,37,40,41].

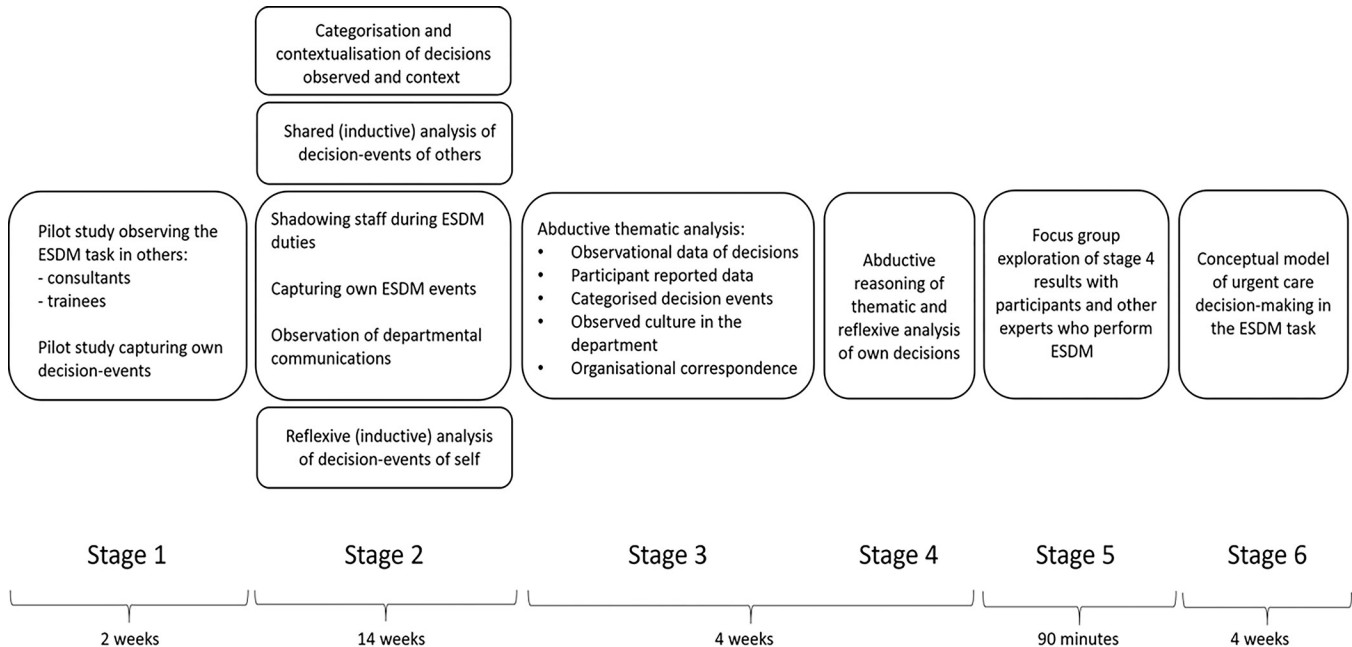

**Fig 1. Process of multimethod data collection and analysis.** The timeline of the research is presented according to the stages of data collection and analysis. ESDM: Early senior decision-maker.

The research complied with the tenets of the Declaration of Helsinki and was approved by the Institutional Review Board at NHS Scotland Regional Ethical Committee and the Health Research Authority, UK. Informed consent was obtained from each participant.

The study had six stages carried out over a three-month time horizon as shown in Fig 1. Findings were iteratively analysed during the data collection process.

## Study setting and participants

The study was based in the acute medical department (AMU) of a large, university teaching hospital in the UK. It served a mixed urban and rural population of >400, 000. The service was for (public) National Health Service care only. The spectrum of illnesses encountered was in keeping with other centres in the UK. As the study was performed during the UK's COVID-19 pandemic (December 2020 to April 2021), the hospital functioned with temporary area for patients in whom COVID-19 was suspected or confirmed (off-site, not observed).

The AMU provided urgent evaluation and care for persons with acute internal medical complaints at all time of the day (e.g., suspected stroke, sepsis, unstable ischaemic heart disease). It received between 45–65 patients daily, with most arriving between 1000-2000hrs. Approximately 70% of patients attended directly from the community following a clinical evaluation with the rest transferring from the Emergency Department. Two senior physicians were present on the unit from 0800-1800hrs to evaluate patients within a few hours of arrival and provide ongoing care of established patients. This reduced to one consultant until 2100hrs, and none overnight. Visiting physicians from other medical departments (gastroenterology, cardiology, respiratory, and elderly care) provided advisory services from 1000-1100hrs and 1700-1800hrs daily. Three to eight trainee physicians (at various stages of training) supported the in-patient and AEC facilities daily according to activity patterns–the most from mid-afternoon to mid-evening, the fewest 2200-0800hrs. The in-patient area had a

patient to nurse ratio of 3:1 with the ability to temporarily deliver critical care in the event of patient decline.

There were two areas for delivering care: an ambulatory emergency care (AEC) facility (urgent out-patient care within 24hrs) and a co-located in-patient area. The AEC facility had clinical space to allow for simultaneous evaluation of seven patients, and comfortable waiting space for a further eight with a companion. The AEC was designed to provide care for physiologically stable patients who required urgent evaluation, but for whom hospital admission was unlikely to be necessary (e.g., venous thromboembolism, minor stroke). It was open from 0800-2300hrs daily after which time, any remaining patients would be transferred to the in-patient area, an alternative ward, or discharged. Patients could attend the AEC over a number of consecutive days for further investigation and care. The in-patient area consisted of 30 Level 1 hospital beds and functioned 24hrs per day. Patients placed here were physiologically unstable, and/or frail, and likely to require hospital admission. Most patients attending (70–80%) required overnight admission. The ability to transfer patients to other parts of the hospital was limited each day by consistently high hospital occupancy levels (>90%).

Participants included the lead author, five consultant (senior) physicians (one female), and three trainee physicians with >4 years training in adult medicine (one female). The lead author had 11 years of experience as a consultant physician specialising in acute internal medicine. Consultant participants ranged in their experience of delivering acute internal medical in the location from five to 16 years. Two trainees were in the early stages of acute internal medical consultant training, the third was in their final year of consultant training in intensive care medicine. The acute medicine trainees had worked exclusively in the department for 48hrs per week for >3months. The intensive care trainee had one-month of 48hr per week experience of the department. The lead author was employed as consultant physician in the organisation for a period of three months. She had previously worked on the site with four of the consultant participants. We assumed consultants to be peer-recognised experts in the early senior decision-making (ESDM) role, which is consistent with assumptions of expertise in the literature. We assumed trainees to be experts in training.

## Data collection

Observation periods were chosen via convenience sampling. The lead author spent three months in the department collecting and analysing data for between four to eight hours every day until 2300hrs at the latest. Her own decision-making shifts were six to eight-hours long and occurred once per week. The lead author recorded her own cognitive processes, including the sequence of events, emotions, and sensations during and/or immediately upon completion of a telephone referral to minimise recall bias. When observing other participants' decision-making referral calls, she took timed notes of dialogue and actions. Immediately post-referral, participants were interviewed face-to-face by the lead author in the private space where the call was taken. Interview notes were made by hand to avoid accidental recording of confidential patient information.

Interviews were subject to the available time following the referral conversation. Interviews followed a loose structure which started by asking participants to recall their decision-making moments during the referral–their awareness of decision-making, emotions/sensations, actions, and timing of decisions. In particular she asked about the moments decision occurred to them, how they emerged, any questions they recalled asking themselves, and how they determined satisfaction with their final decision. Participant recall was cross-referenced with the lead author's notes for clarity on the timing of recalled events. Each interview lasted between 3 to 5minutes–dictated by further referrals calls (requiring immediate answering) and

other urgent clinical discussions required. Decision events were categorised according to the classifications in Box 1 by the lead author and the observed participants during the interviews [42].

> ## Box 1. Decision categories (adapted from Klein et al. [42])
>
> • **Option selection** from externally presented options
>
> • **Deliberation** and rational analysis of multiple options between colleagues or consciously by the decision-maker
>
> • **Constructed, novel creations** based on knowledge of previously seen or shared solutions
>
> • **Procedural** application of a pre-determine rule
>
> • **Analogue** use of another situation seen or heard about before (applied in isolation and not incorporated in a newly created solution)
>
> • **Prototype** a standard approach to previously/often encountered dilemmas, when cases seem to merge into one 'pattern' e.g., chest pain/suspected heart attack

## Data analysis

Data analysis was via the Gioia method of thematic analysis—an iterative process that began as the data was being collected and continued once collection was complete [43,44]. This method supported the use of first-hand accounts and observations to inductively identify first order concepts that captured regularities observed in early decisions about patient allocations across multiple decision events and multiple participants. The lead author performed reflexive analysis of her own decision moments to identify first order concepts that typified her decision-making [35]. First order concepts were then identified in the participant observations and interview data to triangulate the autoethnographic ones [30,34]. Iterative abstraction over the first order concepts created second-order themes via the emergent, inductive enquiries of Grounded Theory and Bracketing [35,43–45]; the latter being a process which brought rigor to the analysis of autoethnographic accounts by abducting upon alternative explanations for the author's own observations and actions [35]. Second order themes represented first order concepts collated into distinct behaviour patterns observed in the expert participants' decision events. Finally, second order themes were abstracted upon to create the aggregate dimensions of early expert decision-making as they relate to decision theory.

A single focus group discussion with consultants practicing ESDM provided a final stage of analysis and internal validation of findings. The findings of the thematic analysis were shared with all consultant staff working in department (n = 10). All were invited to attend an online focus group led by the lead author and/or share their thoughts via private communication. The focus group was attended by two of the observed consultants, and two non-observed consultants (one female). Both non-observed consultants had 2years of experience delivering acute medical care on the site–one being recently qualified as a consultant, the other having over five-years of experience as a consultant at another site. The focus group was an open

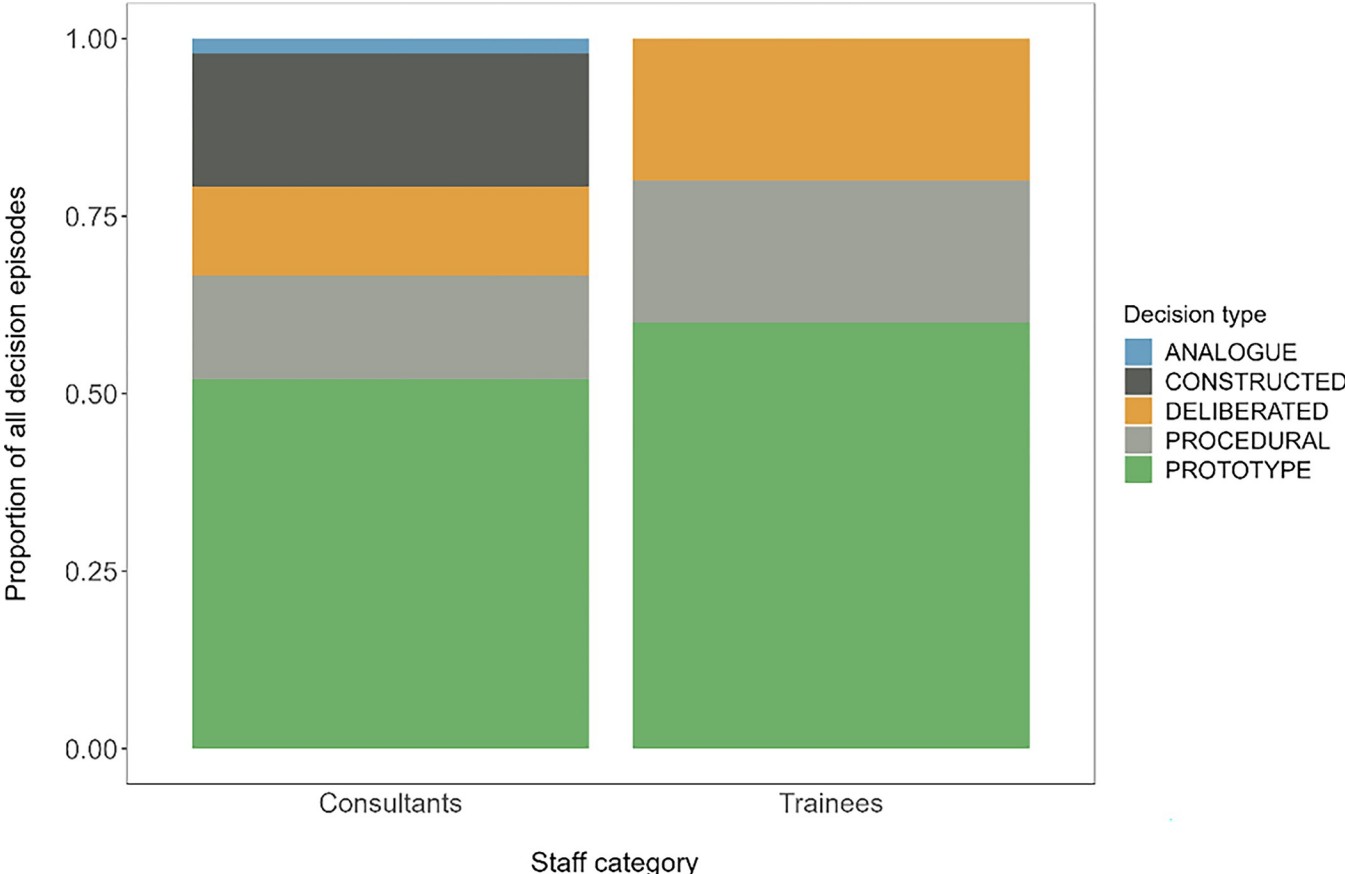

**Fig 2. Decision types observed.** Decision type used by staff are presented as a proportion of the total decisions observed in that staff category: n = 47 in consultants, n = 20 in trainees. Prototype solutions were common amongst both staff groups but consultants employed a greater variety of decision styles. Procedural decisions exclusively concerned decisions to divert patients with suspected COVID-19 infection to an alternative department.

dialogue structured around a single question–"Do the findings describe your experiences of remote, early expert decisions in acute medical care?". Each participant was invited to provide their thoughts on the how ESDM was characterised in the thematic analysis, if they disagreed with any aspects, or if they felt any important factors had not been covered. The discussion lasted 1-hour due to the time constraints of participants.

**Findings.** Sixty-seven decision events were studied: n = 41 for observed consultants, n = 6 the lead author, and n = 20 for trainees. A tabulated summary of the decision episodes may be found at DOI: 10.15129/6af9033c-b8bc-4629-93ec-e4fc8c921611. Consultant showed the greatest variety in decision types. Both groups predominantly using prototype solutions as shown in Fig 2, but only consultants employed constructive (novel) decision-making.

## Thematic analysis

First order concepts, derived from directly observed events and participant descriptions are shown in Fig 3 and serve as the foundation of this analysis. The aggregate dimensions in the figure show that the second order themes were manifestations of the movement between a deliberate, spontaneous, and automatic use of fast thinking supported by conscious deliberation to determine suitability of the solutions generated via fast thinking. The rest of this section provides supporting evidence. To illustrate the findings, we start by presenting an example

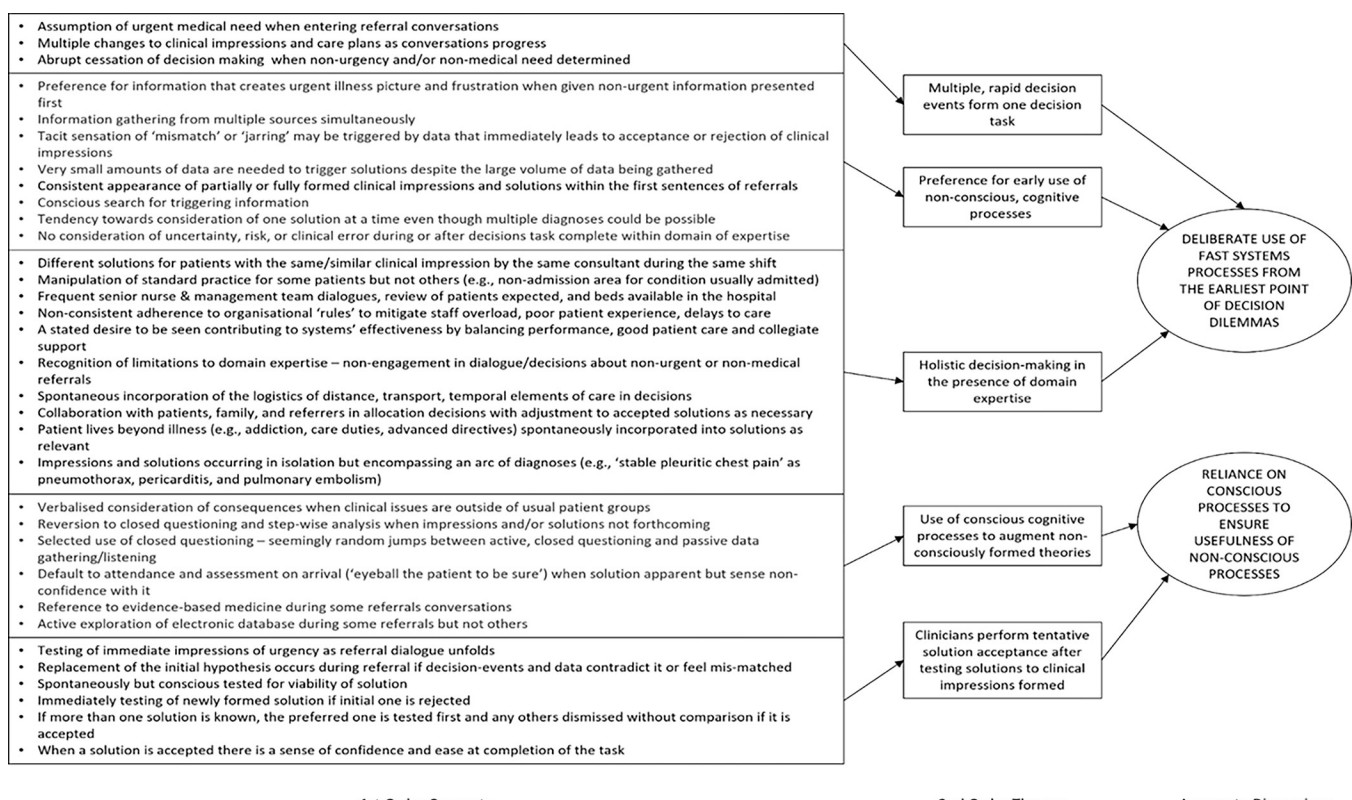

**Fig 3. Thematic analysis of expert decision-making in the early senior decision-maker task.** First order concepts emerge directly from observed behaviour and participant descriptions. These are analysed via an iterative process of conceptual and categorical analysis until second order themes emerge. These themes are categorised in aggregate dimensions which reveal the split between fast and slow thinking employed in expert early decision events.

vignette of the lead authors decision-making during the study. The findings are presented in the first-person format in keeping with autoethnographic tradition.

## Example

The referral phone rings. I'm aware of the sensation of panic and dread about how busy the day is going to be. I answer it, introducing myself and my role.

Paramedic: "Hi doctor, this is [xxx] one of the paramedic crew in [xxx] today. Can I talk to you about a patient?"

I sense anxiety about how busy the day is going to be and look around at the department and check the time for reassurance about capacity in the department.

Me: "Sure. Can you give me a name and a hospital number or date of birth?"

I am logging on to the electronic patient record as I know this will give me access to extra information that may help my decision-making. I sense impatience to decide quickly and move on to my other duties.

Paramedic: "We're with a 42-year-old woman. We were called out to see her because she was having chest pain. . ."

I have an immediate and spontaneous feeling of relief. I have no conscious awareness of a specific diagnosis but a concept or theme with which to filter all information through—'chest pain'. Attendance at the urgent out-patient clinic immediately occurs to me but I don't want to offer this. The patients age seems to trigger a sense of non-urgency but I have a pressing urge to focus on getting more information about this patient to check this initial solution.

The paramedic starts to describe the particulars of the symptoms that alerted the patient. I am listening but simultaneously scanning the summary of her medical records. The paramedic narrative is convoluted. It switches between old symptoms, new symptoms and past medical history. The narrative is interspersed with polite conversation making it difficult to focus on the decision task. I feel my attention move more towards the patient records but moments of dialogue catch my attention: "ECG normal", "10 minutes", "pain-free". I am aware of consciously looking for information that disputes my hypothesis of non-urgency.

I know to look for recent hospital attendances, previously listed conditions, and medication that might offer clues that I am wrong. Data from the conversation and the computer screen makes it feel like information is flowing rapidly into the front of my head. Relevant pieces become trapped there waiting to be used. As pieces of non-relevant data enter and I am conscious of trying to actively 'unknow' them and prevent them from settling there; of having too much information that is non-useful. I instantly know which data is irrelevant for my decision as I hear/see it.

I find a recent hospital attendance and scan through it. The text triggers a sensation of matching with the paramedic description. I feel more relaxed, but not fully. The letters/phrases 'ACS', 'PE', and 'Dissection' spontaneously appear as typed text floating in the front part of my head. I am aware that these diagnoses are my only immediate concern. I mention the previous attendance to the paramedic as I rescan the letter specifically searching for evidence of their exclusion from her diagnosis the last time. As I am reading, "ACS" and "PE" disappear. The word "Dissection" remains. I feel more relaxed still, but not completely. I immediately know I need to see a recent chest X-ray. I find one, look at it for a couple of seconds and feel confident. No need for urgent attendance, but out-patient investigations planned.

## Intentional application of fast thinking

Impressions of urgency or non-urgency surfaced instantly within the initial sentences of a referral conversation. Descriptions of the acute symptoms were used to frame the decision-making episode and rapidly appraise all subsequent information. In the example given, the framing device of 'chest pain' was used to (non-consciously) process the next piece of information offered—the patient's age. As I heard the age, it matched a non-urgent pattern. I was conscious of the potential for this to be a 'lazy' heuristic at the time (young person, low possibility of concern) and considered the risk of bias in this hypothesis formation. I intentionally sought additional information to explore the hypothesis further, testing the potential for bias.

This framing approach was consistent throughout all my decision-events. I would actively try to 'unknow' non-acute information presented at the start of a conversation when information to inform a frame was not forthcoming. I had awareness of appraising fragments of information via these frameworks, categorising data as relevant or non-relevant as it appeared. Information processed incorporated psychosocial concerns, for example alcohol addiction. Categorisation was sometimes consciously done; at other time I had what felt like tacit knowledge of relevancy.

All consultants observed used a framing device in the same way. All stated a preference for acute information early into a conversation and exhibited impatience when appropriate information was not available quickly or lacked clarity. Of particular concern was early knowledge of a patient's background information as it could introduce bias and distort their framing. Like me, they used frames to process subsequent information and evolve hypotheses. Consultants hypotheses focused upon excluding key urgent diagnoses rather than trying to make specific diagnosis.

Biopsychosocial information about the patient was also applied to consultants' frames, but were excluded if clinical concern was great. For example, when both the GP and a patient with

chest pain stated a preference for non-admission (due to carer commitments), this social information was dismissed as non-relevant at that moment because the patient had an abnormal heart tracing. The consultant explained *"the minute I heard the story, I thought 'needs admission'"*. In other instances, psychosocial data was highly relevant—e.g., the patient's distance from the hospital when considering the availability of resources. This ability to rapidly move between individual clinical particulars, the landscape of the hospital system, and non-clinical aspects of care was a notable feature of consultant decision episodes and not observed in trainees. Trainees did not specifically describe early hypotheses or framing, but two of them were observed conducting referrals in a style that suggested they did apply both techniques.

Trainees described early hypothesis formation but their experiences and uses of them were different from consultants. Pattern recognition was a present, but they had fewer patterns in their mental repository, and fewer solutions. They asked more questions than consultants, and spent time trying to generate specific diagnoses with their referrers. When asked about this, two of them explained (independently) that they gathered as much data as possible before arrival to diagnose early and prepare investigations in advance. Trainees were not observed factoring non-clinical information into their decision-making unless specifically asked to by the referrer.

## Key data triggering and emotional affect

Certain pieces of information triggered immediate and confident decisions. Triggering data was usually information that was key to making a final decision about care–a satisfying 'Eureka' moment in the task. In the example given, I initially felt a sense of relief upon hearing the patient's age. However, this was a noticeably different sensation to the easiness I experienced towards the end of the event. Upon analysis, I realised this was a primitive response— the relief from a fear about an overwhelmed department as the patient was unlikely to require admission. As the study continued, I became more aware of the difference between confident ease and the sensations related to my fears of an overcrowded department.

Triggering key data occurred for all staff decision-making, but easiness and confidence was only exhibited in consultants. I frequently observed these moments in participants. Posture change was common when consultants were confident they had found their solution. Physical actions would often be repeated in the same person during different decision episodes, Post-event interview confirmed alignment with confident solution appearance and the physical, and verbal cues observed.

Trainees recognised key data but lacked confidence in their decisions. Observed moments of their key data triggering coincided my own data-triggering moments during referrals conversations. Trainees' body language did not have the same features of relaxation or confidence as consultants. They appeared to struggle with accepting solutions which spontaneously occurred to them, continued to gather information, and made positive efforts to diagnose patients during referrals.

Confidence was noticeably absent when consultants attempted to make decisions outside of domains of expertise. Despite an organisational rule to divert all COVID-19 patients to another department, two consultant participants attempted to apply their usual framing approach when COVID was raised in the referral. They explained this was to reduce demand upon colleagues in the COVID-19 department. One consultant also felt that it enhanced the AMU team's reputation as 'good' decision-makers to exclude COVID-19 early. These referral episodes were lengthy, deliberative, and sometimes involved other members of the team in the deliberation. The consultants who did this described a sense of unease at trying to intuit whether COVID-19 was or was not present after the event. They felt uneasy with decisions

made. They would discuss the decision with me after the episode had ended and would express regret if they chose not divert to the infection control area.

Although emotional affect was a key component of expert decision-making, emotions were also influenced by the decision environment. Overcrowding (occupancy levels beyond the bedded area capacity) presented a safety issue for patients, and a poor experience of care. They overwhelmed and distressed staff. Fears of overcrowding were frequently expressed by consultants:

*"When the pager goes off, I have this immediate dread, like, I know this is another referral and I think 'S***! What is this? Can we cope?' Or is that just me? [laughs]"* Lead author

"No that's me too, I immediately think the patient will need to be admitted" Consultant 1

*"You do get this feeling of panic, but it disappears as soon as you start taking the call"* Consultant 2

Consultants openly discussed allocating more patient to the AEC when overcrowding was present or threatened. I also altered some decisions to prevent when overcrowding was present or possible. These moments were not accompanied by the sensation of easy confidence about individual patient outcomes but felt right for safety goals in that moment. There was consensus amongst us that they were moments of good decision-making.

## Rational analysis to sense-check fast thinking

Fast thinking was complemented by rational analysis (slow thinking) to validate hypotheses and solutions. Rational analyses could be: a conscious effort to make sense of information by gathering more data, a series of closed questions to force the delivery of information in a way that facilitated rapid decision-making, or a falsification test of a solution's suitability (for example, establishing the patient's distance from the hospital to access resources in time). Falsification checks ('sense checks') could be consideration of the likely outcome of a presented solution, but could also be an emotional check that the solution 'felt' right.

The use of rational analysis to complement non-conscious decisions is apparent in the breakdown of a single decision episodes. Decision episodes consisted of multiple spontaneous, rapid decision-events about the nature of illness, urgency, and resource needs which consultants were not always aware of performing. Fig 4 provides an example for a non-admission decision in a patient with alcohol addiction. Although a logical flow can be discerned, this was not wholly apparent to me during the episode. I aware of consciously deliberating some aspects but not others even though they influenced the final decision.

Consultants explained that solutions to decision episodes would appear to them in isolation. They would then automatically and consciously perform a mental check for their suitability. This was consistent with my own experiences. If the first solution that appeared was found to be viable, then no alternatives were considered. If non-viable, another solitary solution would immediately appear to be checked. This pattern would repeat until an acceptable solution emerged. Emotional affect was commonly used to distinguish between heuristics and intuited knowledge when sense-checking. For example, in the case of in Fig 4, I used an emotional check when discerning between a decision based upon tacit knowledge of alcohol addiction or a convenient heuristic that gave me control over resource demand. In contrast, Trainees mostly described generating multiple solutions and then deliberating between them unless there was a clear clinical need for admission (e.g., physiological instability).

**A conceptual model of remote clinical decision-making.** We transformed the results of the thematic analysis into a conceptual model of ESDM by experts for acute internal medical

*Is this immediately life-threatening?*
*(speed on onset, cause of low levels)*

*Does this need intravenous replacement ?*
*(level of phosphate, recent treatment)*

*Can replacement be delivered before the out-patient facility closes?*
*(duration of infusions, resource capacity, transport logistics)*

*Will the patient agree to remain in hospital overnight to complete the treatment?*
*(previous non-compliance, addiction problems, assumptions about behaviours, concerns for resource waste)*

*What will be the impact on the blood level?*
*(effectiveness of intravenous therapy if non-compliant, risk to patient health, alternative strategies)*

*What is the long-term prognosis?*
*(likelihood of rapid recurrence, treatment strategies, futility of urgent treatment)*

**Fig 4. Example of multiple decisions events within a single referral episode.** Example of decision-events recalled in a patient with alcohol addiction, history of discharge against clinician advice, and a low phosphate level. The knowledge required to answer them are shown in parentheses. Decisions occurred in rapid succession. Although shown in a logical sequence this was not how they were experienced.

populations. The interplay between fast and slow cognitive processing is summarised below and presented in a graphical form (Fig 5). An acute clinical framework is established from the urgent symptoms fed to the decision-maker from the external environment (the referrer). The decision-maker filters all information through this framework. As information is processed, hypotheses emerge and tacitly known solutions become instantly available. As all decisions lie within the wider context of the department, and the healthcare system (the decision environment), solutions that occur have to be tested in the acute illness framework and the context of the environment. This requires the focus of decisions to move back and forth between the individual patient and the wider healthcare system. Once a solution has been found to be both

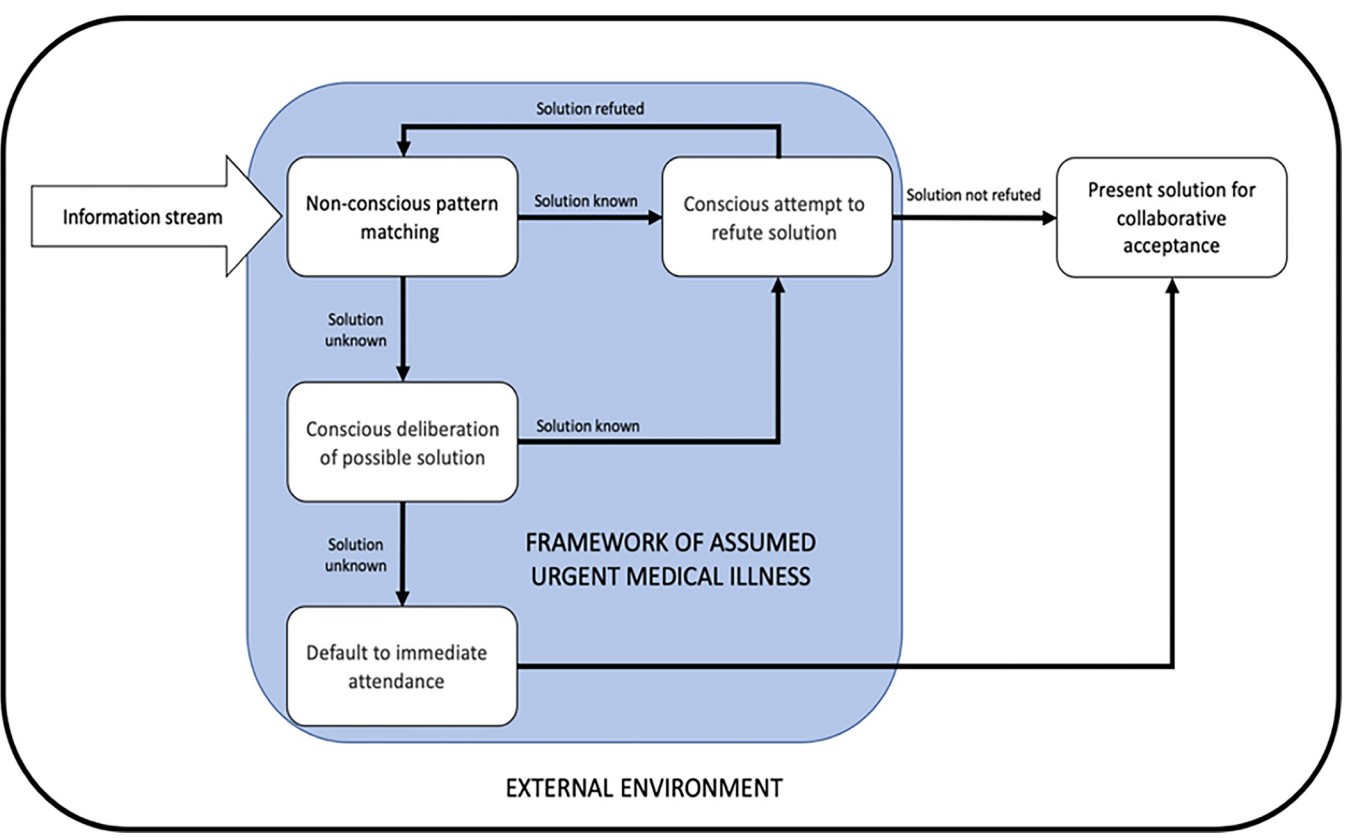

**Fig 5. Conceptual model for expert remote allocation decisions.** Initial information about the urgent health decline generates a framework within which all subsequent information is appraised. This framework exists within the urgent care environment which varies from moment-to-moment. Patterns of information are non-consciously matched to known patterns or amalgamations of known patterns with solutions to address them tacitly known. Patterns and solutions are consciously tested for suitability as they appear before acceptance/rejection.

sufficient and satisfactory, it is accepted as contextually correct. The lightning speed of the entire process is possible via the appearance of key data present within the information stream.

## Discussion

The purpose of the research was to understand how experts perform remote decision-making for acute internal medical populations. It was prompted by absence of knowledge about how clinicians make decisions outside of diagnosing. We found that experts in acute internal medicine rely heavily on intuitive decision-making in early allocation tasks. They intentionally apply fast thinking processes to generate hypotheses patients' and system needs, and useful solutions without the need to generate specific diagnoses. Experts clinicians sense-check the products of non-conscious thought via rational analysis.

Our study augments existing experimental studies of clinicians by revealing that intuitive decision-making is a frequently applied mode of fast thinking by clinical experts in real-world clinical-operational scenarios. Intuitive expertise draws on mental patterns developed via experiential learning [26], although it is sometimes mischaracterised as lightning fast analysis that is "frozen into habit" [25,46]. With intuitive decision-making, the triggering of mental patterns spontaneously and automatically manifests accompanying solutions that may be prototypes, adapted/combined prototypes, or entirely novel solutions [25,47]. Solutions are context-dependent, holistic, and useful for that moment though not necessarily optimal [25,26].

This movement between the particulars and the landscape is a key component of intuitive expertise as is the role of emotional affect [25,26]. Our study reveals that expert clinicians move deftly between the particulars and the wide picture. It also reveals that they are attuned to the differences between primitive and intuitive emotional affect–aware of the influence of the former and creating utility from the latter. While previous clinician studies have hinted at intuition (e.g., Schmidt and Boshuizen's 'illness scripts' [48]), subsequent research has not explicitly explored its role in judgement, intuitive creativity, nor decision-making beyond the clinical realm as we have [15,21].

Our findings corroborate the argument that intuitive decision-making is prevalent among experts across different domains [23,25,28]. While research into forms of fast thinking has revealed the flaws of relying upon what Sinclair terms "low effort mental heuristics" [25], there is consensus among authoritative academics that the form of fast thinking employed in expert intuition differs significantly [23,26,28]. Expert intuitive decision-making and problem solving —where domain-specific knowledge accumulated via experiential learning facilitates rapid pattern matching aimed at specific goals–has been demonstrated in the time-sensitive, high-stakes environments such as fire-fighting and military combat [49–51]. Similar intuitive decision-making processes have also been observed in experts in business and chess [52,53]. Our findings align with the Naturalistic Decision Making framework [25,54,55] and support Simon's concept of 'satisficing' in rapid expert decision-making—the acceptance of contextually sufficient, rather than optimal, solutions [56].

## Limitations

Our study and results have limitations which are important to acknowledge. We were able to triangulate our lead author's reflexive findings with the observed behaviours of and collaboration with expert staff, however, there was no scope for a second researcher to observe and interview. The relative novelty of early expert decision-making also prevented triangulation via observation at an alternative site and our sample size was necessarily small given the nature of the methodology. These final points may limit the generalisability of our findings. We argue that the apprenticeship model of clinical training in medicine, and the mode of urgent care delivery in the UK may be sufficiently comparable to urgent care expertise in other universal health care systems.

We identified three key forms of bias that our methodology was designed to mitigate, but nonetheless remain a limitation. Social desirability bias is an important consideration given the nature of participants' work [57]. As senior clinicians, there is likely to be desire to present themselves to as the pinnacle of professionalism and competency, with altruistic goals. Honesty about how and why ESDM decisions are made (e.g., influenced by fear or exhaustion) may conflict with this presentation. Our methodology was intentionally chosen to limit this as much as possible with a researcher who was familiar, had professional solidarity, and was candid in presenting her own analysis of experiences, but we accept the it is unlikely to have been completely excluded. Our lead author may have been similarly influenced. Recall bias was mitigated as described, but the subconscious nature of decision-making may have influenced participants to present a step-wise rationalisation of decision processes after the fact rather than the processes as actually experienced. Finally, our lead author's analysis of her own decision-events may have introduced researcher bias into the interviews via reaction to participant responses.

## Foundational knowledge for research

Our findings present useful knowledge for future research of clinical and operational decision-making. Firstly, they provide information to plan research evaluating the cost-effectiveness of

an ESDM service–something that would benefit healthcare systems in and beyond the UK. The findings suggest that a controlled experimental study would be challenged in adequately reproducing the variations of information, resources, patients' needs, and the dynamic goal-seeking observed. This complexity would require a long study time-horizon to adequately capture the contextual variations in observational research. Attempts to reproduce ESDM in a healthcare system via computer simulation modelling are likely to be computationally exhaustive. These are not reasons to avoid researching its value, but the enhanced knowledge that our findings deliver advise what may and may not be possible to include when planning research.

Secondly, the findings may be of value to researchers who seek to augment clinician decision-making via artificially intelligent systems [58,59]. This is an area of research too comprehensive to cover in this discussion, and beyond the scope of our work, however, it is clearly of interest to healthcare leaders internationally [58,60]. Developers of decision support systems will require knowledge of the influences and goals of clinical experts.

Finally, our work highlights the inadequacy of previous research designs which assume that clinician experts and non-experts apply the same styles of fast thinking [29]. The appearance of intuitive processes, and the intentional application of some heuristics, and the use of rational analyses to move between them was apparent in our study. Future research into clinical decision-making should consider all forms of fast thinking in their methodological planning, and specifically explore the role, development, and limitations of intuitive decision-making.

## Conclusion

This work reveals the use of intuitive decision-making in clinical experts tasked with combining clinical and operational care decisions. The parallels with Naturalistic Decision Making seen in experts of other high stakes domains suggest that intuitive decision-making may be more prevalent in medicine that previously appreciated. Whilst an optimal strategy for human decision-making in this context, intuitive decision-making is not a strategy that delivers optimal solutions. However, better knowledge of its uses and limitations will aid training and development of future clinicians. This enhanced understanding of the use of intuitive decision-making in clinicians also facilitates knowledge of how artificial intelligence systems may be helpful for the type of decisions clinicians face in the modern healthcare setting. Our findings also have implications for research of expertise in other domains of medical practice where the role of intuition is currently underrecognized.

## Acknowledgments

The authors wish to thank Dr Viktor Dörfler at the Department of Management Science, Strathclyde Business School, UK for his guidance on the topic of intuitive expert decision-making.

## Author Contributions

**Conceptualization:** Nicola Irvine.

**Formal analysis:** Nicola Irvine.

**Investigation:** Nicola Irvine.

**Methodology:** Nicola Irvine.

**Project administration:** Nicola Irvine.

**Supervision:** Robert Van Der Meer, Itamar Megiddo.

**Validation:** Nicola Irvine.

**Writing – original draft:** Nicola Irvine.

**Writing – review & editing:** Nicola Irvine, Robert Van Der Meer, Itamar Megiddo.

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
