## [Decision Letter · Decision Letter 0]

30 Jul 2024

PONE-D-24-02400Expert decision-making in clinicians: An auto-analytic ethnographic study of operational decision-making in urgent carePLOS ONE

Dear Dr. Irvine,

Thank you for submitting your manuscript to PLOS ONE. After careful consideration, we feel that it has merit but does not fully meet PLOS ONE’s publication criteria as it currently stands. Therefore, we invite you to submit a revised version of the manuscript that addresses the points raised during the review process.

We look forward to receiving your revised manuscript.

Kind regards,

Asli Suner Karakulah, PhD

Academic Editor

PLOS ONE

Additional Editor Comments:

Your manuscript has been reviewed and requires modifications prior to making a decision. The comments of the reviewers are included at the bottom of this letter. Reviewers indicated that introduction, methods and discussion sections should be improved. We would be glad to consider a substantial revision of your work, where the reviewer’s comments will be carefully addressed one by one.

Reviewers' comments:

Reviewer's Responses to Questions

**Comments to the Author**

1. Is the manuscript technically sound, and do the data support the conclusions?

Reviewer #1: Yes

Reviewer #2: Yes

2. Has the statistical analysis been performed appropriately and rigorously? 

Reviewer #1: I Don't Know

Reviewer #2: Yes

3. Have the authors made all data underlying the findings in their manuscript fully available?

Reviewer #1: Yes

Reviewer #2: Yes

4. Is the manuscript presented in an intelligible fashion and written in standard English?

Reviewer #1: Yes

Reviewer #2: Yes

5. Review Comments to the Author

Reviewer #1: Dear Author,

Congratulation for conducting this study. I found your study valuable.

Please consider these comments:

Abstract:

1. It is important to briefly describe the setting of study since it influences readers understanding of the study.

2. As your context is specific, I think your study is focused Ethnography.

Methods:

1. Triangulation in method of data gathering, data sources and investigators is necessary. Please declare them as your limitations if they are not committed in your study.

2. Setting of the ward including number of beds, physicians, private or public, exact duration of spending time in the ward for study is necessary.

3. Diverse background of participants should be pointed out by more details.

4. You should have thick description in fact you should explain about relationships and events in addition to physical environment.

5. Detailed explanations about interviews are needed including time, being face-to-face and etc.

6. Was data analysis according to inductive or deductive methods?

7. Describe about considerations of rigor and also triangulation in analysis.

8. Please describe that data collection and data analysis happened simultaneously.

9. Conducting focus group should be described by more details including number of FGs, participants, time, questions, etc.

10. Please describe about participants characteristics.

Discussion:

Please explain about limitations of your work like possibility of social desirability bias or other biases.

Reviewer #2: Overall, this paper significantly contributes to our understanding of expert decision-making in urgent care settings. Addressing the above suggestions could further enhance its impact and clarity. It offers a fresh look at intuitive decision-making in expert clinicians within urgent care, a subject that's not been deeply explored before. Focusing on the cognitive processes behind early expert decision-making, especially in telephone triage, addresses a noticeable gap in current research. Combining analytic autoethnography with ethnographic observation is an innovative method that adds depth to the collected data. This dual approach provides a richer understanding of decision-making processes from both a personal and observational perspective, boosting the study's credibility. Creating a grounded theoretical model of expert decision-making using the Gioia methodology is an inventive way to categorize decision types and processes. This model has potential as a basis for future research and training programs.

The manuscript can be further enhanced if the authors consider the following comments:

1- Clarify the methodological details, particularly how the Gioia method was applied, to help readers better understand the thematic analysis and model development process.

2- Provide a broader contextualization by comparing the findings with decision-making processes in other high-stakes domains, strengthening the argument that intuitive decision-making is a universal skill among experts.

3-Discuss the study's limitations in more detail, including the small sample size and potential bias from the lead author’s dual role. Address the generalizability of the findings to other settings and specialties.

4- Expand on future research directions, particularly in developing training programs or AI systems, providing specific suggestions for experimental studies or simulations to build on these findings.

6. PLOS authors have the option to publish the peer review history of their article (what does this mean?). If published, this will include your full peer review and any attached files.

Reviewer #1: No

Reviewer #2: No

---

## [Author Response · Author response to Decision Letter 0]

11 Sep 2024

We thank the reviewers for their very helpful and constructive feedback on our article. We also thank them for supportive comments about the importance of research in this field and the methodology employed.

We have reviewed the comments and agree with all points made. We have amended our manuscript to reflect the recommendations.

Please note the line referenced refer to the tracked changes manuscript to allow comparison with previous submission.

Reviewer #1: 

Abstract:

1. It is important to briefly describe the setting of study since it influences readers understanding of the study.

We have updated the original manuscript to include a brief description of the urgent care facility, setting, and population (lines 29-30).

2. As your context is specific, I think your study is focused Ethnography.

We have updated the description of methods in the abstract to incorporate the reviewers recommendations of the term ‘Focused Ethnography’ (line 27-28). As a crucial component of the study, the analytic autoethnographic component is also included.

Methods:

1. Triangulation in method of data gathering, data sources and investigators is necessary. Please declare them as your limitations if they are not committed in your study.

A limitations section has been added to the manuscript. This explains the limits of the triangulation that was possible. Lines 590-592

2. Setting of the ward including number of beds, physicians, private or public, exact duration of spending time in the ward for study is necessary.

3. Diverse background of participants should be pointed out by more details.

4. You should have thick description in fact you should explain about relationships and events in addition to physical environment.

We have provided a richer description of the ward setting as advised. This includes the resources, types of patients, relationships, and events pertinent to the study Lines 153 – 196

The duration of time spent has been added to the data collection section at Lines 214-217 

Participants descriptions have been updated to provide more details Lines 198-206

5. Detailed explanations about interviews are needed including time, being face-to-face and etc.

A fuller description of the particulars of the interviews and notes taken has been added Lines 217-238

6. Was data analysis according to inductive or deductive methods?

Description of the Gioia Methodology of analysis has been updated to specify inductive analysis Lines 251-266.

7. Describe about considerations of rigor and also triangulation in analysis.

Methodology section has been updated to improve descriptions of rigor in the method and sources for triangulation. Limitations in triangulation added to end of discussion as described above Lines 120-139

8. Please describe that data collection and data analysis happened simultaneously.

Manuscript adjusted to explain simultaneous collection and analysis of data Line 147

9. Conducting focus group should be described by more details including number of FGs, participants, time, questions, etc.

10. Please describe about participants characteristics.

Description of the focus group has been updated to include more detail as recommended including the participant characteristics Lines 268-285

Discussion:

Please explain about limitations of your work like possibility of social desirability bias or other biases.

Line 588-614 Limitations section added and include biases as recommended by the reviewer

Reviewer #2: 

The manuscript can be further enhanced if the authors consider the following comments:

1- Clarify the methodological details, particularly how the Gioia method was applied, to help readers better understand the thematic analysis and model development process.

Lines 248-266 Methodology section updated as recommended

2- Provide a broader contextualization by comparing the findings with decision-making processes in other high-stakes domains, strengthening the argument that intuitive decision-making is a universal skill among experts.

Lines 572-586 Expansion on discussion of intuitive expertise in other domains

3-Discuss the study's limitations in more detail, including the small sample size and potential bias from the lead author’s dual role. Address the generalizability of the findings to other settings and specialties.

Line 588-614 Limitations section added and include biases as recommended by the reviewer

4- Expand on future research directions, particularly in developing training programs or AI systems, providing specific suggestions for experimental studies or simulations to build on these findings.

We thank the reviewer for this suggestion. Indeed, we did initially consider including more detail about how our work could inform future research into AI systems for decision support in this domain. However, on further reading we realised it is an expansive topic in which we are to advise upon re: methodology of future studies. We acknowledge our limitations in being able to expand upon this further in the final section (Line 629-633)

---

## [Decision Letter · Decision Letter 1]

25 Sep 2024

Expert decision-making in clinicians: An auto-analytic ethnographic study of operational decision-making in urgent care

PONE-D-24-02400R1

Dear Dr. Irvine,

We’re pleased to inform you that your manuscript has been judged scientifically suitable for publication and will be formally accepted for publication once it meets all outstanding technical requirements.

Kind regards,

Asli Suner Karakulah, PhD

Academic Editor

PLOS ONE

Additional Editor Comments (optional):

The authors addressed the reviewers' concerns and substantially improved the content of the manuscript. So, based on my own assessment as an academic editor, no further revisions are required and the manuscript can be accepted in its current form.

Reviewers' comments:

Reviewer's Responses to Questions

**Comments to the Author**

1. If the authors have adequately addressed your comments raised in a previous round of review and you feel that this manuscript is now acceptable for publication, you may indicate that here to bypass the “Comments to the Author” section, enter your conflict of interest statement in the “Confidential to Editor” section, and submit your "Accept" recommendation.

Reviewer #2: All comments have been addressed

2. Is the manuscript technically sound, and do the data support the conclusions?

Reviewer #2: Yes

3. Has the statistical analysis been performed appropriately and rigorously? 

Reviewer #2: Yes

4. Have the authors made all data underlying the findings in their manuscript fully available?

Reviewer #2: Yes

5. Is the manuscript presented in an intelligible fashion and written in standard English?

Reviewer #2: Yes

6. Review Comments to the Author

Reviewer #2: (No Response)

7. PLOS authors have the option to publish the peer review history of their article (what does this mean?). If published, this will include your full peer review and any attached files.

Reviewer #2: No

---

## [Editor Report · Acceptance letter]

30 Sep 2024

PONE-D-24-02400R1 

PLOS ONE

Dear Dr. Irvine, 

I'm pleased to inform you that your manuscript has been deemed suitable for publication in PLOS ONE. Congratulations! Your manuscript is now being handed over to our production team.

Kind regards, 

on behalf of

Dr. Asli Suner Karakulah 

Academic Editor

PLOS ONE